# The Influence of Storage Temperature and Packaging Technology on the Durability of Ready-to-Eat Preservative-Free Meat Bars with Dried Plasma

**DOI:** 10.3390/foods12234372

**Published:** 2023-12-04

**Authors:** Paweł Pniewski, Krzysztof Anusz, Ireneusz Białobrzewski, Martyna Puchalska, Michał Tracz, Radosław Kożuszek, Jan Wiśniewski, Joanna Zarzyńska, Agnieszka Jackowska-Tracz

**Affiliations:** 1Department of Food Hygiene and Public Health Protection, Institute of Veterinary Medicine, Warsaw University of Life Science, Nowoursynowska 159, 02-776 Warsaw, Polandjoanna_zarzynska@sggw.edu.pl (J.Z.); 2Department of Systems Engineering, University of Warmia and Mazury in Olsztyn, Heweliusza 14, 10-724 Olsztyn, Poland; 3Facility of Audiovisual Arts, Institute of Journalism and Social Communication, University of Wrocław, Joliot-Curie 15, 50-383 Wrocław, Poland

**Keywords:** shelf life, RTE meat bar, blood plasma, MAP, vacuum packaging, Aerobic Plate Count, lactic acid bacteria, yeast and mould, *L. monocytogenes*, growth model

## Abstract

Food business operators must include the results of shelf life testing in their HACCP plan. Ready-to-eat preservative-free meat products enriched with blood plasma are an unfathomable area of research in food safety. We tested modified atmosphere (80% N_2_ and 20% CO_2_) and vacuum packaged RTE preservative-free baked and smoked pork bars with dried blood plasma for Aerobic Plate Count, yeast and mould, lactic acid bacteria, *Staphylococcus aureus*, Enterobacteriaceae, *Escherichia coli,* and *Campylobacter* spp., and the presence of *Listeria monocytogenes* and *Salmonella* spp. during storage (temperatures from 4 to 34 °C) up to 35 days after production. The obtained data on the count of individual groups of microorganisms were subjected to analysis of variance (ANOVA) and statistically tested (Student’s *t*-test with the Bonferroni correction); for temperatures at which there were statistically significant differences and high numerical variability, the trend of changes in bacterial counts were visualised using mathematical modelling. The results show that the optimal storage conditions are refrigerated temperatures (up to 8 °C) for two weeks. At higher temperatures, food spoilage occurred due to the growth of aerobic bacteria, lactic acid bacteria, yeast, and mould. The MAP packaging method was more conducive to spoilage of the bars, especially in temperatures over 8 °C.

## 1. Introduction

The increased demand for ready-to-eat (RTE) food caused the need to research [1] food characterised by reduced calorific value associated with lowering the fat content and increasing the protein content in the product composition. Many food producers make business analyses to use relatively low-budget raw materials that will be safe and will not affect food’s safety and sensory quality. The ideal raw material to be used to achieve the newly set consumer goals seems to be animal blood plasma, which is a rich source of protein with a balanced amino acid composition and is also a functional element, apart from its building function that strengthens the structure of the product thanks to its water-binding and emulsifying properties of lipid compounds [2]. Blood plasma constitutes approx. 65% of the blood volume in a domestic pig, which is approx. 3 L of valuable raw material (5% of the pig’s live weight). Estimating the risk of using plasma as a food additive can significantly improve the ergonomics of processing animal products. The mean concentrations of constituents in porcine plasma are total solids 8.5%, ash 1.7%, protein 6.1%, non-protein nitrogen 0.2%, and fat 0.2%; while mean mineral concentrations of porcine plasma are sodium 2734 ppm, potassium 312 ppm, magnesium 14 ppm, calcium 116 ppm, iron 9 ppm, and chloride 2912 ppm. In addition, a mean of 7.8 for pH values is reported [3]. Hence, the high pH of the dried plasma with a slightly alkaline reaction causes an increase in the pH of the RTE meat bar, which may increase the risk of the growth of food spoilage and pathogenic microorganisms affecting the durability of food products.

Often, during the slaughter of animals, blood is classified as an animal by-product under Regulation (EU) No. 1069/2009 of the Parliament and the Council due to the lack of appropriate technology to obtain it in a possibly sterile manner, avoiding contact between the plasma and the surface of the animal’s skin [4]. Appropriate post-slaughter treatment of the blood and further steps of obtaining dried plasma, including mixing the blood with an anticoagulant (potassium or sodium citrate), centrifugation, decolourisation with hydrochloric acid, vacuum concentration, and spray drying, may reduce losses of animal raw material in the food industry [5].

Due to the growing trend for the production of RTE food with dietary features (low calorific value due to high protein and low fat content) and low-processed food (minimal product preservation, especially chemical preservation related to the use of nitrites), it is necessary to analyse the microbiological safety of newly created products. The scientific evidence gathered from successful experimental trials validates the HACCP system. Each food business operator (FBO) is obliged to ensure the compliance of the product with the microbiological criteria set out in the Regulation of the European Parliament and of the Council No. 2073/2005 and provide scientific evidence during storage, confirming the safety of products placed on the domestic and international market under the Regulation of the European Parliament and of the Council No. 852/2004 [6,7].

The RTE, high-protein, preservative-free meat bar with dried plasma considered in this study is an innovative new product. There is no literature on the shelf life and durability of this type of food. For this reason, it was necessary to undertake a wide range of microbiological studies, taking into account food spoilage bacteria (APC, LAB, yeast and mould, *S. aureus*, Enterobacteriaceae) and pathogens (*L. monocytogenes*, *Salmonella* spp., *E. coli*, *Campylobacter* spp.) considering different packaging technologies and storage temperatures.

RTE foods pose a particular hazard due to *L. monocytogenes* occurrence. Their ability to grow at low temperatures (below 4 °C), under aerobic and anaerobic conditions and in MAP (modified atmosphere packaging) products, make them a significant concern for the food industry. They necessitate control measures along the food chain to prevent listeriosis, a severe disease which, according to EFSA reports, is responsible for 1900 hospitalisations per year [8,9,10]. A previous report says that up to 13.5% (*n* = 334) of RTE pork products in China showed the presence of *L. monocytogenes* [11]. Other studies involving 1049 roasted pork sausages showed that 1.8% did not meet the limit for the absence of *L. monocytogenes* in 25 g of product [12].

In addition, it has been confirmed that the storage temperature is paramount to maintaining the appropriate biological quality of food until shelf life [13]. Another study using Monte Carlo prediction based on historical data showed that controlling the refrigerated storage temperature of vacuum-packed long shelf life RTE products reduces the risk of *L. monocytogenes* infection for a healthy population (risk level 10^−6^) and high-risk groups (risk level 10^−4^) by 80% [14]. Other work compared three packaging methods of sliced RTE pork meats regarding the presence of *L. monocytogenes.* They reported the most frequent presence in products packed in atmospheric air (8.9%, *n* = 101), less in vacuum-packed products (4.0%, *n* = 200), and least often in MAP products (3.2%, *n* = 820) [15].

Furthermore, it has been described that meat products packaged with access to atmospheric air spoiled mainly due to the growth of yeast and mould, and vacuum-packed products spoiled due to the growth of LAB (*Enterococcus*, *Lactococcus*, *Lactobacillus*, *Leuconostoc*, *Pediococcus*); however, we are referring to products cured, i.e., those with the addition of preservatives in the form of sodium nitrite [16]. They also emphasised that product formulation, storage temperature, and packaging technology are crucial in uncured RTE product technology. Among the hazards to uncured MAP or VP (vacuum packaging) products, sporadic contamination of the product with Enterobacteriaceae and *Aeromonas* spp. was mentioned. Still, the main reinforcing factors were the permeability of the packaging material to atmospheric oxygen, the initial level of contamination, pH, and storage temperature.

Mathematical modelling is an integral part of the scientific evidence used to guide the response of food safety risk managers to these systems. Food systems can primarily benefit from continuously improved information technology and data from the food and related sectors. Consequently, there is a continued interest in developing new solutions that help harness the potential of information technology in the food sector [17].

The presence of microorganisms in food is essential for the quality and durability of the food. Using mathematical modelling obtained from quantitative studies on microbial populations, FBO may predict the effect of food factors, such as water activity (a_w_), pH, storage conditions, temperature, and relative humidity, on microbial growth in food [18].

It has been confirmed that prognostic models provide results at least 1000 times faster than challenge tests, which consist of estimating the shelf life of a product under the same conditions as during the distribution or storage of food based on the count of microorganisms, determining the acceptable level or the level of deterioration [19].

Most models developed in predictive microbiology require a nonlinear regression method, such as the Baranyi and Roberts growth model or the modified Gompertz equation, which describes changes in bacterial concentration over time. As food experiments are complex and labour intensive, simplified experimental models are often used in practice where it is possible to keep the most critical factors under control. Over the past few years, a growing trend has been to develop predictive models from food data to obtain more accurate predictions [20,21].

Kinetic models can predict the level of contamination of a given microorganism, thus determining the onset of impending risk of infection or poisoning. These models are calculated based on growth or death response rates [21]. The most commonly used kinetic secondary models include the square root model, the gamma model, and the polynomial function [22].

The Gompertz, Weibull, Davey, and Ratkowsky square root models and the McMeekin polynomial secondary model have been used to develop a secondary model of *E. coli* and *C. jejuni* behaviour in milk [23]. Based on probabilistic models, predictive models and dose–response models using the Weibull model, the Davey model, and the root-mean-square error, it has been confirmed that there is a low risk of developing campylobacteriosis due to the consumption of contaminated minced meat [24].

A basic simulation model estimated zero probability of *Salmonella* spp. and *S. aureus* contamination from consuming RTE egg products (e.g., rolled omelette). Primary growth models for both strains in the egg yolk were developed based on the Baranyi model. Secondary models were developed as a function of temperature for lag phase duration (LPD) and maximum specific growth rate (μ_max_) using Davey polynomial, secondary polynomial, and square root models, respectively [25].

The researchers, using mathematical modelling, were able to predict an APC count in the bacterial contamination load range from 1.7 to 7.8 log cfu/cm^2^ with a prediction error of 0.752 log cfu/cm^2^ based on the results of fluorescent fingerprint spectrophotometry of 30 lean beef meat slices each of Australian and Japanese cattle stored aerobically at 15 °C [26].

Considering all these reports, we evaluated the effect of two different reduced oxygen packaging (ROP) methods and various storage temperatures on the durability (use-by date) of preservative-free RTE pork bars with porcine blood plasma. We assessed product durability by testing microorganisms according to Process Hygiene Criteria (PHC) such as Aerobic Plate Count (APC), Enterobacteriaceae, yeast and mould, *Staphylococcus aureus*, lactic acid bacteria (LAB), and food safety criteria (FSC), such as *E. coli*, *Salmonella* spp., *Campylobacter* spp., and *L. monocytogenes*. Based on the collected data, four-parameter mathematical models were used to describe the trend of changes in microbial abundance over time in tested temperature ranges, paralleling the Baranyi and Gompertz growth models.

## 2. Materials and Methods

### 2.1. Product Sample

We used 246 RTE meat bars from 3 independent production batches throughout the study. All bars were manufactured in one medium-sized facility (capacity ca. 10,000 kg weekly) located east of the Masovian district.

The main meat raw material for the RTE preservative-free meat bars with pork plasma production was chilled boneless pork ham (parts of *mm. semimembranosus*, *quadriceps*, *biceps*, *semitendinosus*, and *gluteus*) obtained six hours after the slaughter of large white pigs from Polish farms. Other supplementary raw materials were table salt (sodium chloride), ground black pepper (*Piper nigrum*), Combivit at a concentration of 5 g/kg of the meat filling (composed of lactate, E327; sodium acetate, E262; dioxide, E551; SOVIT, Warsaw, Poland), and dried pork plasma.

The nutritional value of dried plasma (AProPork™, Essentia Protein Solutions, Gråsten, Denmark) was 1364 kJ in 100 g and was composed of fat 2% (including saturated fatty acids 0.8%), carbohydrates 2% (including sugars 1%), protein 74.9%, and salt 11.9%. The mineral components of the dried plasma were (in 100 g of the product) calcium (108 mg), phosphorus (1530 mg), sodium (4750 mg), potassium (651 mg), magnesium (24 mg), and chlorides (3220 mg). The amino acid composition of dried plasma was as follows (in 100 g of product): alanine (4.19 g), arginine (4.57 g), aspartic acid (6.97 g), cystine (1.72 g), glutamic acid (10.04 g), glycine (2.92 g), histidine (2.7 g), isoleucine (2.85 g), leucine (6.44 g), lysine (6.82 g), methionine (0.75 g), phenylalanine (2.7 g), proline (3.6 g), serine (4.27 g), threonine (4.49 g), tryptophan (1.05 g), tyrosine (3.22 g), and valine (5.62 g).

A complete flowchart of the meat bar’s production and the technological parameters at the control points are shown in Figure 1. The desired organoleptic characteristics of the final product in the form of a meat bar included a rough, non-smooth, dry surface with acceptable wrinkles, light to dark brown colour, and an aroma typical of finely ground sausages (a mesh diameter of 5 mm). FBO carried out reduced oxygen packaging (MAP or VP) using a double-chamber packaging machine (C500, MULTIVAC, Wolfertschwenden, Germany).

### 2.2. Storage

On the day the product was delivered to the laboratory (day zero of the test, corresponding to the day the production of the bars ended), the bars were separated into control and test samples and tested or placed in incubators, respectively.

Control samples constituted the samples tested on day zero (equivalent to the end of production date). The microbiological tests of the control group included determining the total count of APC, LAB, *E. coli*, Enterobacteriaceae, *S. aureus*, *Campylobacter* spp., and yeast and mould. The presence of *Salmonella* spp. and *L. monocytogenes* in 25 g of the sample was also determined. Moreover, physicochemical parameters, such as a_w_ and pH, were measured.

Test samples were classified as MAP or VP according to the reduced oxygen packing technology. After giving each bar a unique code, an equal number of bars were placed in thermal incubators with different target temperatures for the MAP group (4, 8, 12, 16, 20, 24, 30, and 34 °C) and VP group (4, 6, 8, 12, 16, 20, and 24 °C). The measurement points during storage of the MAP bars were days 1, 3, 7, 10, 14, 17, 21, 24, 28, and 35, whereas the measurement points during the storage of VP bars were days 3, 7, 12, 14, 16, and 21. Definitions of treatments and sampling schemes are shown in Table 1.

On measurement point, two samples of each MAP and VP group for every researched temperature were removed from each incubator and used for the assessment of the count or presence of examined microorganisms, which included PHC (APC, Enterobacteriaceae, yeast and mould, LAB, *S. aureus*) and FSC (*E.coli*, *Salmonella* spp., *Campylobacter* spp., and *L. monocytogenes*). In addition, on days 7, 14, 21, 28, and 35 for the MAP group and 3, 7, 12, 14, 16, and 21 for the VP group, the pH and a_w_ of RTE meat bars were measured. A simplified diagram of the laboratory experiment is shown in Figure 2.

### 2.3. Physicochemical Examination

Parallel to the microbiological examinations of the meat bars, a_w_ and pH measurements were carried out during storage on days 3, 7, 12, 14, 16, and 21 for the VP group and on days 7, 14, 21, 28, and 35 for the MAP group.

We determine the a_w_ of the meat bar samples using the AQUALAB 4TE water activity meter (METER Group Inc., Pullman, WA, USA) according to validation standard with reference number ISO 18787:2017 [27].

The prepared 1/10 dilutions were tested in a SevenCompact™ pH meter S210 (Mettler-Toledo International Inc., Greifensee, Switzerland).

### 2.4. Numerical Analysis of Microbiological Indicators in RTE Meat Bars

The assessment of the total count of APC, Enterobacteriaceae, yeast and mould, *E. coli*, *Campylobacter* spp., and *S. aureus* in RTE meat bars after storage on measurement days is presented in Table 1. The assessment was performed by the TEMPO^®^ Automatic System for Counting Quality Indicators (bioMérieux SA, Marcy-l’Étoile, France).

On subsequent measurement days, bars were aseptically opened and homogenised (25 g of products/225 mL BPW/2 min, forming a 1:10 dilution of the meat sample). Next, 1 mL of the sample was added to TEMPO^®^ bottles with a dried culture medium and filled with 3 mL of sterile water to create a 1/40 concentration. In order to dissolve the culture medium and homogenate thoroughly, it was vortexed for 10 s (Vortex Mixer VX-200, Labnet International, Edison, NJ, USA). For results that fell outside the 1:40 dilution cutoff, further dilutions of 1/400, 1/4000, 1/40,000, 1/400,000, and 1/4,000,000 were prepared.

After hermetically filling, cards were incubated (CO_2_ incubator type CB-150, Binder GmbH, Tuttlingen, Germany) to determine the count of LAB (30 °C/48 h), APC (30 °C/24 h), Enterobacteriaceae (35 °C/24 h), yeast and mould (25 °C/72 h), *E. coli* (37 °C/24 h), *S. aureus* (37 °C/24 h), and (*Campylobacter* spp. (42 °C/48 h).

After a defined incubation period (measurement days), the cards were read, and the count of specific microbes was calculated based on the number and size of the positive microtubes (fluorescence or lack of fluorescence) and the statistical method of analysis (Bernoulli diagram to estimate the most probable number). Results were presented as the logarithm of the number of colony-forming units per gram of product (log cfu/g). The arithmetic mean with the standard deviation from three repetitions was calculated and thus presented. For data for which the numerical variability was sufficiently high (α = 0.05), we developed mathematical models of microbial growth/survival.

### 2.5. Immunoassay Confirmation of the Presence of L. monocytogenes and Salmonella spp. in Meat Bars

To assess the presence of *L. monocytogenes* and *Salmonella* spp. in a meat bar after the incubation period in question at nine various storage temperatures (4, 6, 8, 12, 16, 20, 24, 30, and 34 °C), we used a mini VIDAS^®^ Compact Immunoassay Analyzer (bioMérieux SA, Marcy-l’Étoile, France) [28,29].

On each measurement day, the meat bars were opened aseptically, and 25 g samples of each were placed in two bags: the first for *L. monocytogenes* and the second for *Salmonella* spp. identification. Next, 225 mL of Half-Fraser Broth and 0.5 mL of LMX Supplement were added to 25 g of the bar prior to 2 min homogenisation. The exact amount of Buffered Peptone Water and one tablet of *Salmonella* Supplement was added to the sample before homogenisation for *Salmonella* spp. identification.

After homogenisation, samples intended for *L. monocytogenes* identification were incubated at 37 °C for 26 h, and samples intended for *Salmonella* spp. identification were incubated at 42 °C for 24 h. Next, 2 mL of each test sample were transferred to microtubes to heat at 95–100 °C for 5 min in a water bath.

After the heating stage, 0.25 mL of *L. monocytogenes* identification suspension was overlaid on the sump of the LMX strap previously put in the analyser with a pipette internally coated with specific antibodies against *L. monocytogenes.* The same operation applies to the sample for *Salmonella* spp. identification, differing in volume (0.5 mL of suspension) and the types of straps (SPT straps).

The mini VIDAS^®^ analyser delivers the sample examination results after 50 min for *Salmonella* spp. and 60 min for *L. monocytogenes* identification using the enzyme immunofluorescence ELFA technique (Enzyme-Linked Fluorescent Assay).

### 2.6. Bacterial Kinetic Models

We assumed that the bacterial kinetics model proposed to explain the investigated growth/survival processes carried out during storage at temperatures of 4, 8, 12, 16, 20, 24, 30, and 34 °C is described by a two-term functional relationship in Equation (1), where A, B, and K are the constants in the model; pH is the reaction of the environment; a_w_ is water activity; X is the population of microorganisms (log cfu/g); r is the growth constant (1/h); γ_wa_ is the coefficient describing the effect of water activity on bacterial growth; and γ_pH_ is the coefficient describing the effect of pH on bacterial growth:(1)$$\frac{dX}{dt}=\;r\cdot \gamma_{pH}\cdot \gamma_{wa}\cdot X\cdot \left(1-\frac{X}{K}\right)-\gamma_{pH}\cdot \gamma_{wa}\cdot \frac{B\cdot X^{2}}{A^{2}+X^{2}}$$

We examined the location of the γ_pH_ coefficients (Equation (3)) in the model, taking into account the influence of pH on the count of microorganisms, and γ_wa_ (Equation (2)). Reckoning with the impact of a_w_ on the count of microorganisms ensures its best fit to the experimental data. The form of the γ_pH_ and γ_wa_ coefficients is analogous to what was proposed when determining the influence of temperature on the biotechnological processes studied [30]. The minimum, optimal, and maximum values of a_w_ and pH are 0.87, 0.95, and 0.99 and 4.2, 6.7, and 9, respectively.
(2)$$\gamma_{wa}=\frac{\left(aw-aw_{max}\right){\left(aw-aw_{min}\right)}^{2}}{\left(aw_{opt}-aw_{min}\right)\left\{\left(aw_{opt}-aw_{min}\right)\left(aw-aw_{opt}\right)-\left(aw_{opt}-aw_{max}\right)\left(aw_{opt}+aw_{min}-2aw\right)\right\}}$$
(3)$$\gamma_{pH}=\frac{\left(pH-pH_{max}\right){\left(pH-pH_{min}\right)}^{2}}{\left(pH_{opt}-pH_{min}\right)\left\{\left(pH_{opt}-pH_{min}\right)\left(pH-pH_{opt}\right)-\left(pH_{opt}-pH_{max}\right)\left(pH_{opt}+pH_{min}-2pH\right)\right\}}$$

The instantaneous values of pH(t) and a_w_(t) were determined by searching based on experimental data for unknown values of the coefficients r_pH_, K_pH_, and r_wa_ and K_wa_, respectively, in differential Equations (4) and (5):(4)$$\frac{dpH}{dt}=\;r_{pH}\cdot pH\cdot \left(1-\frac{pH}{K_{pH}}\right)$$
(5)$$\frac{dwa}{dt}=\;r_{wa}\cdot wa\cdot \left(1-\frac{wa}{K_{wa}}\right)$$

The appropriate algebraic Equations (6) and (7) describing the instantaneous values of pH(t) and wa(t) were used to determine the values of the γ_pH_ and γ_wa_ coefficients to speed up the numerical solution of the model equation.
(6)$$pH\left(t\right)=\frac{pH_{0}\cdot K_{pH}\cdot e^{r_{pH}\cdot t}}{K_{pH}+pH_{o}\cdot \left(e^{r_{pH}\cdot t}-1\right)}$$
(7)$$wa\left(t\right)=\frac{wa_{0}\cdot K_{wa}\cdot e^{r_{wa}\cdot t}}{K_{wa}+wa_{o}\cdot \left(e^{r_{wa}\cdot t}-1\right)}$$

The optimisation procedure, knitro_nlp, implemented in the Knitro 12.4 package (Artelys, Paris, France), was used to determine the values of the searched coefficients r, γ_pH_, γ_wa_, K, A, and B in the model consisting of Equations (1)–(3) and (6)–(7). This package was used in the MATLAB 2021a environment (MathWorks, Natick, MA, USA). The objective function, J_c_, was in the following form:(8)$$Jc\left(model\;coef.\right)=\sum_{i=1}^{n}{\left(\frac{X_{mes}-X_{mod}}{X_{mes}}\right)}^{2}$$

The values of the coefficients sought were determined as a solution to the optimisation problem. Likewise, the same procedure, with the appropriate form of the objective function, was used to determine the unknown values of the coefficients in Equations (4) and (5).

### 2.7. Statistical Analysis

Statistical analysis was performed using Microsoft Excel 2019 (Windows, Redmond, WA, USA) using a one-way analysis of variance (ANOVA).

Multiple means repetitions were performed using the Student’s *t*-test with the Bonferroni correction at a significance level of α = 0.05.

## 3. Results

### 3.1. Storage Tests of Refrigerated RTE Meat Bars

The raw experimental data of the microbial group counts (APC, LAB, *S. aureus*, and yeast and mould) in the samples of RTE meat bars from VP_C and MAP_C groups are presented in Appendix A.

The APC values in the MAP_C bars remained practically the same throughout the entire storage period; they reached 4.23 log cfu/g at 4 °C and 4.43 log cfu/g at 8 °C. VP_C bars showed a similar tendency for the count of aerobic microorganisms to MAP_C bars. However, their values were slightly higher than half a log unit at 5.17, 5.16, and 5.30 log cfu/g at 4 °C, 6 °C, and 8 °C, respectively. ANOVA showed that the effect of storage time on the count of aerobic microorganisms in RTE bars stored at 4, 6, and 8 °C for both MAP_C and VP_C groups was not statistically significant (*p* > 0.01).

LAB count in MAP_C bars showed low numerical variation during storage at 4 °C. The initial value on day 0 was 2.02 log cfu/g, with a maximum on day 17 (2.75 log cfu/g). The largest statistically significant increase (described as “Δ” is the difference between the measurement day and the previous measurement day; *p* < 0.01) in LAB count was observed on day 10 (Δ = 1.61 log cfu/g). In the MAP_C bar at 8 °C, the maximum LAB count was reached on day 35 (3.94 log cfu/g), and the most significant increase in LAB count was on day 10 (Δ = 1.80 log cfu/g). The VP_C bars had a higher initial LAB count on day 0 (3.76 log cfu/g) than the MAP_C bars; however, unlike the MAP_C bars, LAB count in the VP_C bars tended to decrease from day 0, except on day 12, when the maximum LAB count reached 2.92 log cfu/g, 3.07 log cfu/g and 3.83 log cfu/g at 4 °C, 6 °C, and 8 °C, respectively.

*S. aureus* counts in VP_C and MAP_C bars are heterogeneous. MAP_C bars stored at 4 °C showed a decreasing trend in staphylococcal counts, reaching a maximum on day 0 (2.44 log cfu/g) and a minimum on day 10 (0.77 log cfu/g). Statistically significant (*p* < 0.01) increases occurred on day 7 (Δ = 1.04 log cfu/g), 14 (Δ = 0.78 log cfu/g), and 21 (Δ = 1.10 log cfu/g). A similar increase of approximately one log unit was observed in the MAP_C bars at 8 °C on days 7 (Δ = 0.98 log cfu/g), 14 (Δ = 1.93 log cfu/g), and 21 (Δ = 1.01 log cfu/g).

In the VP_C bars, the numerical variability of *S. aureus* count was less dispersed, with 1.12 log cfu/g of the initial count on day 0. At 4 °C, the count of *S. aureus* peaked on day 21 (2.00 log cfu/g), with significant increases recorded on day 12 (Δ = 0.95 log cfu/g) and day 21 (Δ = 0.98 log cfu/g). Similar *S. aureus* behaviour in VP_C bars could be observed during storage at 8 °C; maximum *S. aureus* count on day 21 (2.16 log cfu/g) and significant increases of 1 log unit on day 21 (Δ = 1.05 log cfu/g). *S. aureus* counts in the VP_C bars stored at 6 °C reached their maximum on day 14 (2.20 log cfu/g) and a significant increase (over 0.5 log units) on day 21 (Δ = 0.84 log cfu/g).

In the VP_C and MAP_C bars, the growth of yeast and mould was rarely detected. In MAP_C bars stored at 4 °C and 8 °C, it was detected only on 4 out of 11 measurement days (at 4 °C on days 3, 17, 24, 35 and at 8 °C on days 14, 17, 24, 28) and exceeded 1 log cfu/g only once (MAP_C bars at 8 °C on day 28). The presence of yeast and mould on day 0 was 0.33 log cfu/g in the VP_C bars.

There was no change in the *E. coli* and *Campylobacter* spp. counts in the refrigerated meat bars (VP_C and MAP_C) compared to the initial counts on day 0 (0 log cfu/g). Also, no change in the Enterobacteriaceae count was observed in the VP_C and MAP_C meat bars. The exceptions were results obtained for samples stored at 4 °C (MAP_C, day 3), 6 °C (VP_C, day 21), and 8 °C (VP_C, day 21), for which the values were 0.80 ± 1.13 log cfu/g, 0.77 ± 1.08 log cfu/g, and 0.33 ± 0.47 log cfu/g, respectively.

*Salmonella* spp. and *L. monocytogenes* were not found (not detected in 25 g) in VP_C and MAP_C bars during every measurement day of the storage test.

### 3.2. Storage Tests of RTE Meat Bars Stored under Non-refrigerated Conditions

The raw experimental data of APC, LAB, *S. aureus*, and yeast and mould in the VP_NC and MAP_NC RTE meat bars are shown in Appendix A.

The initial (measured at “Day 0”) APC value in the MAP_NC bars was lower (4.43 log cfu/g) than the initial APC value in the VP_NC bars (5.12 log cfu/g). At non-refrigeration temperatures, there was a statistically significant (*p* < 0.01) increase in APC values in the MAP_NC bars stored above day 17 at 12 °C (Δ = 1.63 log cfu/g) and at 16 °C (Δ = 1.76 log cfu/g). During storage at higher temperatures, the increase in APC values in the MAP_NC bars was observed much earlier: above day 10 at 20 °C (Δ = 1.01 log cfu/g), above day 3 at 24 °C (Δ = 1.30 log cfu/g), above day 1 at 30 °C (Δ = 1.56 log cfu/g), and above day 7 at 34 °C (Δ = 1.33 log cfu/g). Maximum APC values in the MAP_NC bars were reached at the latest on day 28 at 12 °C (6.34 log cfu/g), while at the earliest on day 10 at 30 °C (8.84 log cfu/g) and 34 °C (8.85 log cfu/g). The decrease in the APC value for the MAP_NC bars at 34 °C stored above 10 days (Δ = −1.70 log cfu/g) was statistically significant (*p* < 0.01).

A change in at least more than half a logarithmic unit in the APC values in the VP_NC bars compared to the initial count occurred only at 24 °C on day 7 (Δ = 0.72 log cfu/g) and 12 (Δ = 0.66 log cfu/g). At other non-refrigeration temperatures, the fluctuations of the APC value relative to the initial value remained below half a log unit, reaching a maximum APC value on day 16 at 12 °C (5.58 log cfu/g) and 16 °C (5.50 log cfu/g), day 12 at 20 °C (5.62 log cfu/g), and day 7 at 24 °C (5.84 log cfu/g).

The initial count of LAB in the MAP_NC bars (2.02 log cfu/g) was lower than in the VP_NC bars (3.76 log cfu/g); however, it should be noted that the wide standard deviation of the count of LAB in the VP_NC bars showed the resultant variability. LAB surge in the MAP_NC bars occurs at the latest after 7 days of storage at 12 °C (Δ = 1.08 log cfu/g) and at the earliest on day 1 at 20 °C (Δ = 1.74 log cfu/g), at 24 °C (Δ = 1.55 log cfu/g), at 30 °C (Δ = 2.97 log cfu/g), and at 34 °C (Δ = 2.92 log cfu/g). The lowest peak of LAB count in the MAP_NC bars was on day 35 at 12 °C (5.43 log cfu/g), and the highest peak was on day 24 at 30 °C (8.70 log cfu/g).

The count of LAB in the VP_NC bars was numerically less diversified than in the MAP_NC bars. The increase in LAB count above the initial value was recorded only twice on day 16 at 20 °C and 24 °C, reaching 4.10 log cfu/g and 4.28 log cfu/g, respectively. Statistically significant increases in LAB count (above 1 log unit) were recorded on day 7 at 12 °C (Δ = 1.07 log cfu/g) and 24 °C (Δ = 1.68 log cfu/g), on day 16 at 16 °C (Δ = 1.96 log cfu/g) and 24 °C (Δ = 1.06 log cfu/g) g), and on day 12 at 20 °C (Δ = 1.55 log cfu/g).

The MAP_NC bars initial count of *S. aureus* was 2.44 log cfu/g. Its number showed a downward trend at 12 °C and 16 °C and never exceeded the initial value. At 20 °C, an increase of 2.29 log cfu/g was observed on the third day of storage (reaching a maximum of 3.47 log cfu/g), and then from day ten onwards, the count of staphylococci gradually decreased. Also, at 24 °C, the count of *S. aureus* significantly increased by 1.54 log cfu/g on day 10 (reaching a maximum of 3.31 log cfu/g), subsequently decreasing. At 30 °C, three significant (*p* < 0.01) increase peaks have been seen: on day 3 (Δ = 1.98 log cfu/g), 10 (Δ = 3.08 log cfu/g), and 17 (Δ = 1.90 log cfu/g). A similar trend has been observed in *S. aureus* growth when stored at 34 °C in the MAP_NC bar; the bacteria count increased significantly on day 3 (Δ = 1.88 log cfu/g), day 10 (Δ = 1.10 log cfu/g), and day 17 (Δ = 1.25 log cfu/g).

The initial *S. aureus* count in the VP_NC bars (1.12 log cfu/g) was lower than in the MAP_NC bars (2.44 log cfu/g). The trend in the count of *S. aureus* in the VP_NC bars was essentially constant and exceeded 2 log cfu/g only on day 12 at 12 °C (2.02 log cfu/g), 16 °C (2.01 log cfu/g), and 20 °C (2.18 log cfu/g), and on day 14 at 12 °C (2.19 log cfu/g).

There was no evidence of yeast and mould growth in the MAP_NC bars directly after production (when tested on day 0). The VP_NC bars on day 0 were characterised by a low yeast and mould presence at a level equal to 0.33 log cfu/g. No yeast or mould was found in the VP_NC bars at 16 °C and 24 °C on the measurement days during storage for three weeks. In the VP_NC bars at 12 °C, the only growth was observed on day 14 (1.26 log cfu/g). At 20 °C, yeast and mould were present on days 12, 16, and 21, reaching 1.55 log cfu/g, 1.22 log cfu/g, and 1.47 log cfu/g, respectively.

Different yeast and mould growth dynamics were observed in the MAP_NC bars. The growth started the latest on day 10 at 12 °C (1.00 log cfu/g) and the earliest on day 1 at 24 °C (2.52 log cfu/g) and 30 °C (0.58 log cfu/g). Maximum values of yeast and mould count in the MAP_NC bars were reached on day 17 at 30 °C (5.88 log cfu/g) and 34 °C (6.03 log cfu/g), showing a statistically significant upward trend.

*E. coli* and *Campylobacter* spp. were not found in any MAP_NC and VP_NC bar samples.

The lack of growth throughout the storage test was also shown for the Enterobacteriaceae in the VP_NC bars. In the MAP_NC bars, intermittent growth of Enterobacteriaceae occurred only for 2% of bars (*n* = 198): on day 3 at 16 °C, 24 °C, and 34 °C, with an average count of 1.16 ± 1.65 log cfu/g, 1.12 ± 1.58 log cfu/g, and 1.04 ± 1.47 log cfu/g, respectively, and on day 7 at 16 °C, with an average of 0.43 ± 0.61 log cfu/g.

*Salmonella* spp. and *L. monocytogenes* were not found in any 25 g sample of the VP_NC bars and MAP_NC bars until days 21 and 35 of storage, respectively.

### 3.3. Changes in pH and a_w_ Values in RTE Meat Bars during Storage

The raw experimental data on the changes in the pH and a_w_ values in the samples of RTE meat bars stored at 4, 8, 12, 16, 20, 24, 30, and 34 for the MAP group and 4, 6, 8, 12, 16, 20, and 24 °C for the VP group are shown in Appendix A.

The sensitivity analysis showed no statistically significant changes in pH and a_w_ over time in the VP_C, VP_NC, and MAP_C bars. Statistically significant (*p* < 0.01) changes in pH and a_w_ occurred only for the MAP_NC bars at temperatures as low as 24 °C. However, the most noticeable pH and a_w_ decrease between day zero and the end of the experiment occurred in bars stored at 30 °C and 34 °C. The initial and final pH values were 0.68 and 0.83, respectively, and the difference between those values for a_w_ was 0.037 and 0.049, respectively.

### 3.4. Determination of Mathematical Models of Microbial Growth/Survival in RTE Meat Bars

All possible combinations of the γ_pH_ and γ_wa_ coefficients were tested in the AC model (Equation (1)). The best fit, measured by the value of the coefficient of determination R^2^ (presented in Table 2), was obtained for the form of the model in Equation (9):(9)$$\frac{dX}{dt}=r\cdot \gamma_{pH}\cdot X\cdot \left(1-\frac{X}{K}\right)-\frac{B\cdot X^{2}}{A^{2}+X^{2}}$$

This form of the model shows that a_w_ does not significantly affect the count of bacteria in the conducted research.

The growth/survival of LAB and yeast and mould in MAP bars at non-refrigeration temperatures were performed in a simpler model described by Equation (10).
(10)$$\frac{dX}{dt}=r\cdot \gamma_{pH}\cdot \gamma_{wa}\cdot X\cdot \left(1-\frac{X}{K}\right)$$

For this form of the model, all possible combinations of *γ_pH_* and *γ_wa_* coefficients were also tested, i.e., these coefficients and their product separately. After analysing the value of the coefficient of determination R^2^ calculated for all temperatures, it was found that the introduction of the *γ_pH_* and *γ_wa_* coefficients into the model did not significantly improve its fit; therefore, the final form of the AC bacterial colony behaviour model is in the form of Equation (11):(11)$$\frac{dX}{dt}=r\cdot X\cdot \left(1-\frac{X}{K}\right)$$

Small changes in the R^2^ value as a function of the *r* parameter, as presented in Table 2, Table 3 and Table 4, can be observed for all temperatures, proving the numerical stability of the APC, yeast/mould, and LAB models for the MAP_NC bars.

In the tests, the presence of *Campylobacter* spp., Enterobacteriaceae, and *E. coli* in the MAP_C and MAP_NC bars was practically not found; therefore, the process of determining the coefficients in the model was not carried out for them. The VP_NC and VP_C bars also showed no increase in *Campylobacter* spp., Enterobacteriaceae, and *E. coli* count above the minimum detectable value (10 cfu/g). Moreover, occasional results above 10 cfu/g in the yeast and mould count in the MAP_C, VP_NC, and VP_C bars also provided too little of an elemental database for determining reliable mathematical models.

Low numerical variability of changes in the count of *S. aureus* in the MAP_NC and MAP_C bars, as well as the count of APC, LAB, *S. aureus,* and yeast and mould in the MAP_C, VP_C, and VP_NC bars, causes the curves of the mathematical models to not show fundamental differences between changes in the four theoretical growth/death phases of microorganisms. The nature of the model lines for the microorganisms mentioned above is relatively constant, which means that particular storage conditions (temperature, packaging method) do not significantly affect their growth. Therefore, they were not included in this publication.

We carried out statistical analysis for all the experimental results obtained; nevertheless, for the data on lower storage temperatures, we did not observe statistically significant differences and large variability in the abundance of individual microorganisms or groups of microorganisms, in contrast to the data on higher (but still non-refrigerated) temperatures. As a result of these observations, we applied a mathematical modelling tool to illustrate the nature of the observed phenomena more clearly. The use of statistical analysis and mathematical modelling of some of the experimental data is not contradictory but complementary; ANOVA showed whether there were statistical differences between the results, and mathematical modelling was used to describe the trend of microbial counts in RTE meat bars where the differences are high and significant. Model curves showing the growth phases for APC, yeast/mould, and LAB in the MAP_NC bars are shown in Figure 3.

Using modelling, the predicted lag phase for APC in the MAP_NC bars at 12 °C (Figure 3A) lasts up to 10 days from production. Then, there is a sudden growth phase lasting approximately 10 days, after which growth is inhibited, and the APC level stabilises around 6 log cfu/g. Lactic acid bacteria at this temperature grow logarithmically until day 21, reaching a relatively constant phase of approximately 5 log cfu/g. The increase in yeast and mould becomes noticeable only on around day 14.

The growth curve for LAB in the MAP_NC bars at 16 °C is similar (Figure 3B). A logarithmic (exponential) growth phase is observed until around day 21, after which a stationary phase occurs, oscillating around 5.5 log cfu/g. The growth curve for APC in the MAP_NC bars stored at 16 °C looks completely different. In this case, there is no typical lag, exponential growth, and stationary, and death phases. Only a steady increase is observed from the beginning of storage until the last measurement day, reaching about 7 log cfu/g at the end of the storage period. The growth curve of yeast and mould in the MAP_NC bar at 16 °C is similar to the APC growth curve at 12 °C; the lag phase lasts for the first three days and then starts an exponential growth phase until day 21, followed by the stationary phase at a level of approx. 3.5 log cfu/g.

The APC growth curve modelled for the MAP_NC bar stored at 20 °C (Figure 3C) does not show a lag phase. From the beginning of storage, there is a growth phase lasting until day 28, when it reaches the stationary phase at approximately an APC count of 7.5 log cfu/g. The LAB model for the MAP_NC bars stored at 20 °C has the same course as the APC model line, except for the earlier end of the growth phase on day 14, when it reaches the stationary phase at 6 log cfu/g. The lag phase for the yeast and mould growth in the MAP_NC bar stored at 20 °C lasts until the first day of storage, after which the increase is expected to occur around day 17, when the stationary phase occurs, reaching a level of 3 log cfu/g.

In general, the course of the growth curves for microorganisms in the MAP_NC bars stored at 24 °C (Figure 3D) is similar to the growth curve model for the MAP_NC bars stored at 20 °C, except that the growth phase of APC lasts until day 14 (reaching the constant phase at the level of 8 log cfu/g), while the growth phase of LAB lasts until day 7 (reaching the constant phase at the level of 6.5 log cfu/g). In contrast, the growth phase in the yeast and mould model reaches the stationary phase at the level of 3.5 log cfu/g. The death phase is not predicted in any model regarding the MAP_NC bars stored at 24 °C. More dynamic growth of APC, LAB, and yeast and mould is predicted in the MAP_NC bars stored at 30 °C compared to 24 °C (Figure 3E); they lack the lag phase, and the constant phase is reached on day 5 at the level of 8 log cfu/g, 7.5 log cfu/g, and 3 log cfu/g for the APC, LAB, and yeast and mould models, respectively.

Noticeable changes compared to the previous measurement temperature (30 °C) occur in the models of APC and yeast and mould counts for the MAP_NC bar at 34 °C (Figure 3F). In the APC model, the growth phase lasts until day 10, when it reaches a maximum of approximately 7.5 log cfu/g. This is followed by a five-day stationary phase, after which there is a distinct death phase (starting from day 20 of storage). It is noteworthy that this is the only case of all the models presented in this work when the death phase was observed during the experiment. The model estimated for yeast and mould has a one-day lag phase, followed by a growth phase lasting until day 10, and a stationary phase lasting until the end of the measurement days (at 1.5 log cfu/g). The growth curve of the LAB population in the MAP_NC bars stored at 34 °C has the same course as the growth curve model for the MAP_NC bars stored at 30 °C, but the stationary phase is half the logarithm lower.

## 4. Discussion

The premise of not using preservatives in the production of RTE meat bars with dried blood plasma is a significant challenge for food operators to ensure the product’s safety, especially if FBOs want to store it outside refrigerated counters. Our research has shown that the product tested as a preservative-free RTE meat bar is safer and lasts longer for human consumption during vacuum packaging than modified atmosphere packaging. The average pH of the RTE no-preservative meat bars was 6.1, and the a_w_ was 0.95, which means that the product tested is a food matrix, supporting the growth of *L. monocytogenes* [6]. In our study, significant changes in pH values occurred only for the MAP bars stored at temperatures above 20 °C. We also observed significant a_w_ changes in the MAP bars during storage at temperatures higher than 30 °C, which argues in favour of vacuum packaging of the bars due to a more physicochemically stable environment for background microflora [31].

In our study, organoleptic signs of spoilage occurred as early as the seventh day after production when the MAP bars were stored under unrefrigerated conditions. Organoleptic changes included an exacerbated sour smell, a change in the bar’s texture from elastic to soggy, and the appearance of a yellow and cloudy discharge of meat juice, which was caused mainly by the growth of LAB, mesophilic aerobic bacteria, and yeast and mould, as shown by the results obtained. The most noticeable growths of mould thallus were in the MAP_NC bars at 24 °C and 30 °C, starting from day 7 of storage. The maximum yeast and mould count was detected in the MAP_NC bars stored at 24 °C (4.68 log cfu/g on day 14) and 30 °C (5.88 log cfu/g on day 17).

Aerobic mesophilic microorganisms generally remained at the same level in the VP_C and MAP_C bars. Their initial count was relatively high (approx. 4.5–5 log cfu/g) compared to most RTE products on the market (approx. 3 log cfu/g) [32]. On day zero, the APC values attributed to the bars we tested are also higher compared to cured-cooked meat studies, where the day zero APC count for the VP group was 3.85 log cfu/g (with a max. of 8.12 log cfu/g on day 30) and 3.83 log cfu/g (with a max. of 8.57 log cfu/g on day 30) for the MAP group [33]. This correlation is mainly due to the addition of preservatives (i.e., sodium nitrate), which significantly reduce the contamination of the finished product and increase its safety. In our study, storage in the VP_NC bars slightly supported the growth of mesophilic aerobic microorganisms, but to a much lower extent than in the MAP_NC bars. In the VP_NC bar, there is an increase of less than ten times, while in the MAP_NC bar, on day 14 at 34 °C the level of APCs increased more than ten thousand times. Notably, the level of APCs in the MAP_NC bar stored at 24 °C and 30 °C reached a maximum and remained there until the final storage time (the “end-day”); while at 34 °C, it reached a maximum around day 14 and then gradually decreased. This observation is not correlated with a drastic decrease in pH, as a similar correlation is not observed for the MAP_NC bars at 30 °C. Due to the non-increasing level of APC in the VP_C bars, the optimal consumption date should be up to 14 days.

LAB count in the MAP_C bars stored at 4 °C remained similar throughout the entire storage period; packing in MAP technology and storage in refrigeration conditions did not support LAB growth. In comparison, clear upward trends were seen when the bars were stored at 8 °C and above. The temperature most conducive to LAB growth in the MAP_NC bars was 30 °C. LAB population in the MAP_NC bars stored at 34 °C reached the death phase on day 14 of storage, as shown in Figure 3F. The increase in the count of LAB strongly correlates with the increase in the pH value in the MAP_NC bars, which was first observed at 20 °C and was most noticeable at 30 °C and 34 °C.

The VP technology did not significantly affect LAB count. Moreover, our research confirms the theory that claims that the growth of LAB in VP products had little impact on the deterioration of the product’s organoleptic characteristics [34]. It was claimed [33] that the count of LAB in cured meat products stored at 4 °C VP was 7.64 log cfu/g on day 30 of storage, and the product packed in a modified atmosphere reached 8.34 log cfu/g on day 30 of storage. The discrepancy with [33] is undoubtedly due to using a different meat processing technology involving curing salt, which was not a target in our study.

However, the LAB results presented are consistent with the findings of other authors [35] who modelled the growth of *Lactobacillus plantarum* in cooked chopped vacuum-packed RTE pork stored at various temperatures. It has been shown that the phase of the rapid growth of *L. plantarum* as an exponential function began on day 0 when stored at 16 °C and on day 4 at 10 °C. When storing the product at 4 °C, the growth of *L. plantarum* started on the 24th day. Nevertheless, it has not reached an exceeding value, which causes deterioration of product quality and risk to consumer health. The results obtained regarding the count of LAB during the storage of RTE meat bars are also in line with the conclusions [36]; however, the LAB results regarding the VP_C bars contradict the conclusions [37,38] that claim that the leading group of bacteria that spoil vacuum-packed foods stored under refrigeration are LAB.

The storage of bars packed in ROP technology (VP and MAP) did not significantly support the growth of *S. aureus;* its number remained at a similar level as the initial count (shortly after production) or had slight numerical fluctuations. In our opinion, the initial number of *S. aureus* in the meat bars, despite implementing good production practices and procedures based on HACCP principles at the production facility, was relatively high (approx. 2 log cfu/g). We recorded high levels of *S. aureus* at the production stage, indicating that the product was intraoperatively contaminated. Our conclusions align with the results of other researchers [39] who investigated the occurrence of *S. aureus* in dry cured meat products and found *S. aureus* in 13.4% (*n* = 83) of the products. *S. aureus* was also more prevalent on food contact surfaces during production than immediately after cleaning procedures. Since the food safety criterion is not *S. aureus* but the toxins produced by it, in further research, the sequencing of genes encoding enterotoxins (*seg*, *sei*, *sem*, *sen*, *seo*, *and seu*) from isolates we obtained from stored RTE bars are considered.

The yeast and mould count in the MAP_NC bars increased as the storage temperature increased, reaching a maximum growth rate at 24 °C and 30 °C. We found the presence of yeast and mould in the VP_C and MAP_C bars very rarely. This is consistent with studies of cooked RTE ham [40], during which a marked increase in the count of yeast and mould was noted on day 48 when stored at 2 °C (Δ = 4.3 log cfu/g), day 29 when stored at 8 °C (Δ = 3.2 log cfu/g) and 12 °C (Δ = 1.5 log cfu/g), and day 7 when stored at 15 °C (Δ = 1.8 log cfu/g).

Our study experienced high standard deviations, often due to detecting yeast and mould in one of three replicates. An explanation for this phenomenon could be the survival of yeast and mould cells present in the product due to contamination; for example, at the packaging stage. These results lead to the need to extend research to all stages of the production process, which may result in new solutions to reduce yeast and mould contamination after heat treatment.

The results indicate that RTE bars produced without preservatives with the addition of dried plasma do not pose a risk to consumers regarding infection with *E. coli*, *L. monocytogenes*, *Salmonella* spp., *Campylobacter* spp., and Enterobacteriaceae; we draw this conclusion based on storage tests, which do not indicate the presence or growth of these pathogens during shelf life testing, regardless of employing the ROP technology that the bar was packaged. Nevertheless, to be completely certain of this conclusion, we plan to extend the research presented with a challenge test involving artificial contamination to determine the growth potential of individual pathogens.

The *L. monocytogenes* results are consistent with some reports [33]; however, in the final result for Enterobacteriaceae and *E. coli*, the first group of microorganisms in their research initially obtained a count of 1.25 log cfu/g with max. 5.57 log cfu/g on day 21 (VP group) and 1.4 log cfu/g with max. 3.09 log cfu/g on day 30 (MAP group), while the latter was initially not detected but reached the maximum on day 30 at 4.04 log cfu/g (VP group) and 2.94 log cfu/g (MAP group); no common point can be found compared to our results.

The presented results of Enterobacteriaceae counts align with the previous research [40], where it was concluded that the count of Enterobacteriaceae in RTE cooked ham during storage does not change from the initial count; this phenomenon was true for each of the storage temperatures tested.

We did not detect any *L. monocytogenes* in the 25 g MAP_NC bars, which may be related to the high count of background microflora, especially LAB. Similarly [41], a noticeable correlation of the growth constant of *L. monocytogenes* at different temperatures in the RTE-cooked ham packed in a modified atmosphere was observed, depending on the count of LAB.

Previous researchers [42] successfully used mathematical modelling using the modified Gompertz equation to predict the change in the abundance of APC, coliforms, and LAB in salted napa cabbages during storage at different temperatures (5, 22, and 30 °C). Based on the presented models, it was possible to determine the optimal shelf life for each storage temperature using specific limits for APC (7.7 log cfu/g) and LAB (6.0 log cfu/g). Assuming their limits for APC and LAB, the shelf life of RTE meat bars should be, based on the presented models, 20 days (16 °C, MAP_NC), 10 days (20 °C, MAP_NC), 5 days (24 °C, MAP_NC), and 2 days (30 °C, MAP_NC); in most cases, the main limiting factor was the overpopulation of LAB. Moreover, the conclusions are convergent and confirm that the temperature rise is vital for the growth of food-spoiling bacteria, which is also indicated by our research.

Many authors use the Gompertz model to calculate the use-by date of food products. The modified Gompertz function (after re-parameterisation) is a commonly used empirical one. Despite its widespread use, the Gompertz model has weaknesses, which include non-horizontal course and estimation of the negative lag value of the phase or imprecise expression of the maximum growth rate of the population by the inflexion point of the curve, which suggests that the maximum growth rate may be higher than the estimated growth rate [43].

Based on mathematical modelling, we reached similar conclusions to previous researchers [44] on the changes in the amount of LAB in vacuum-packed cooked meat emulsions packaged in low-O_2_ permeability film at three different temperatures (0 °C, 8 °C, and 15 °C. The models also show a trend of faster LAB growth with increasing storage temperature, reaching the stationary phase on about days 8 (at 15 °C), 14 (at 8 °C), and 30 (at 0 °C).

Adhering to the criteria proposed for RTE food products [33] for determining the optimal shelf life based on APC (no more than 5 log cfu/g) and LAB (no more than 7 log cfu/g) counts, we conclude that RTE no-preservative meat bars should be stored at refrigeration temperatures (up to 8 °C) for a maximum of 14 days. We recorded the longest shelf life for MAP bars stored at 4 °C; for this product type, the shelf life could be extended from 14 to 21 days.

To produce the RTE high-protein meat bars with blood plasma, which will be stable at non-refrigerated temperatures, FBOs must abandon the premise of producing preservative-free products or use alternative methods of microbial reduction, such as physical (high-pressure processing, pulsed light), chemical (antimicrobial agents), and biological (bacteriophages) methods [45,46,47,48,49,50,51,52,53,54,55,56,57,58,59].

The next point of consideration is adjusting the optimal gas mixture in MAP. The mixture of 80% CO_2_ and 20% N_2_ used in the research is the most commonly used MAP gas mixture in the USA [60]. A study [61] proved that with a gradual decrease in the CO_2_ share from 100%, the count of Enterobacteriaceae, *Pseudomonas* spp. and, above all, LAB, increased. Similar conclusions were reached [62], where lower APC and Enterobacteriaceae values were observed in roast duck meat when using 50% CO_2_ than 30% CO_2_ MAP technology; however, no significant differences in LAB counts were observed.

Congenial research was carried out [63], where the impact of various concentrations of gas mixtures on the levels of APC, LAB, and yeast and mould in roasted chicken meat was examined; they found that the most significant bacteriostatic effect offered to samples when packed in gas mixture contained the highest CO_2_ level among the tested groups (40% CO_2_ and 60% N_2_).

The success of active packaging in microbial control may vary depending on other external factors (i.e., storage temperature, ageing duration, initial loadings). Consequently, intelligent packaging devices that detect microbial loads and species prevalence can be cost effective for producers. These indicators can also inform industry stakeholders of the need to take action on microbial levels; this means that it is not right to expect microbial levels to remain below the threshold of acceptability indefinitely; an acceptable level of safety should also be maintained between processing and consumption, and it is up to the food manufacturer to assess the safety of the solutions used [64].

RTE preservative-free meat bars with blood plasma are undoubtedly an innovative product newly created due to consumer demand. Their high protein content, the absence of chemical food preservatives, low production processing, and intention for readiness to eat (without the need for heat treatment before consumption) increase the likelihood of the product spoiling sooner than expected. Therefore, all newly created alternative products require precise shelf life testing under adequate temperature stress.

## 5. Conclusions

Our research concludes that packaging methods and storage conditions are critical to meeting the food safety criterion (FSC) and ensuring pork meat bars’ expected shelf life (FSC and PHC). Considering the results we obtained for all tested groups of microorganisms, the preferred reduced oxygen packaging (ROP) for ready-to-eat preservative-free meat bars with dried plasma is vacuum packing followed by refrigeration storage, optimally at 4 °C (max. 8 °C). The consideration of packaging RTE bars using MAP technology is warranted if the tests are extended to individual gas mixture compositions, assuming the use of CO_2_ concentrations above 20%.

As the durability tests showed, the optimal shelf life for VP bars is 14 days when stored at 4 °C, 12 days at 6 °C, or 10 days at 8 °C. Extending the shelf life of vacuum-packed and refrigerated bars is only possible if additional hurdles (e.g., HPP, pulsed light, antimicrobials) are implemented and scientific evidence of their effectiveness is provided.

The production of additive-free bars requires the implementation of good hygiene practices at the plant to the highest level. Consideration should be given to isolating a microbiological high-risk zone in the production area, which includes the packaging stage, as a control measure for the microbiological hazard of recontamination of the product.

Although we did not detect the presence of *L. monocytogenes* in any 25 g sample, there is no clear evidence if the pork meat bar with dried plasma is a food that does or does not support the growth of *L. monocytogenes*. Taking into account the average pH and aw values of the bars, we assess that this product should be classified, according to Regulation 2073/2005, as food category 1.2 ready-to-eat foods able to support the growth of *L. monocytogenes*, other than those intended for infants and special medical purposes [6], unless there are future scientific reports presenting challenge test data indicating a different classification for this type of food.

## Figures and Tables

**Figure 1 foods-12-04372-f001:**
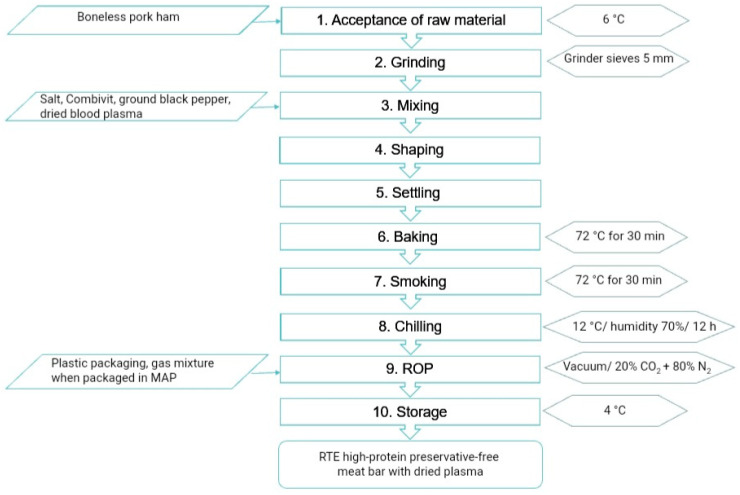
Flowchart of the production process of RTE preservative-free meat bars with dried plasma.

**Figure 2 foods-12-04372-f002:**
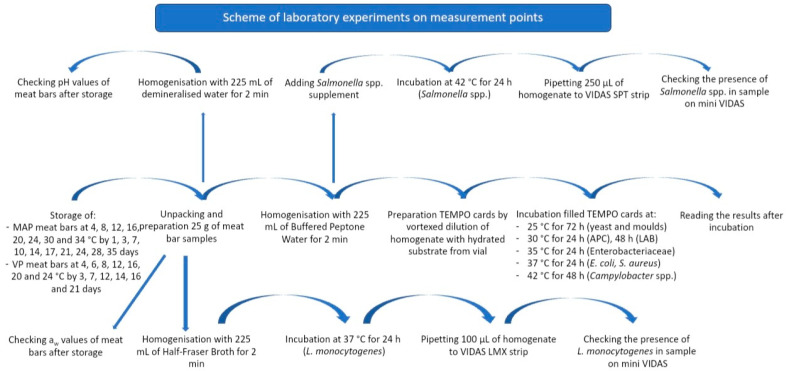
Simplified scheme of the laboratory experiment for storage tests of RTE no-preservative meat bars with dried blood plasma packed in MAP and VP technology.

**Figure 3 foods-12-04372-f003:**
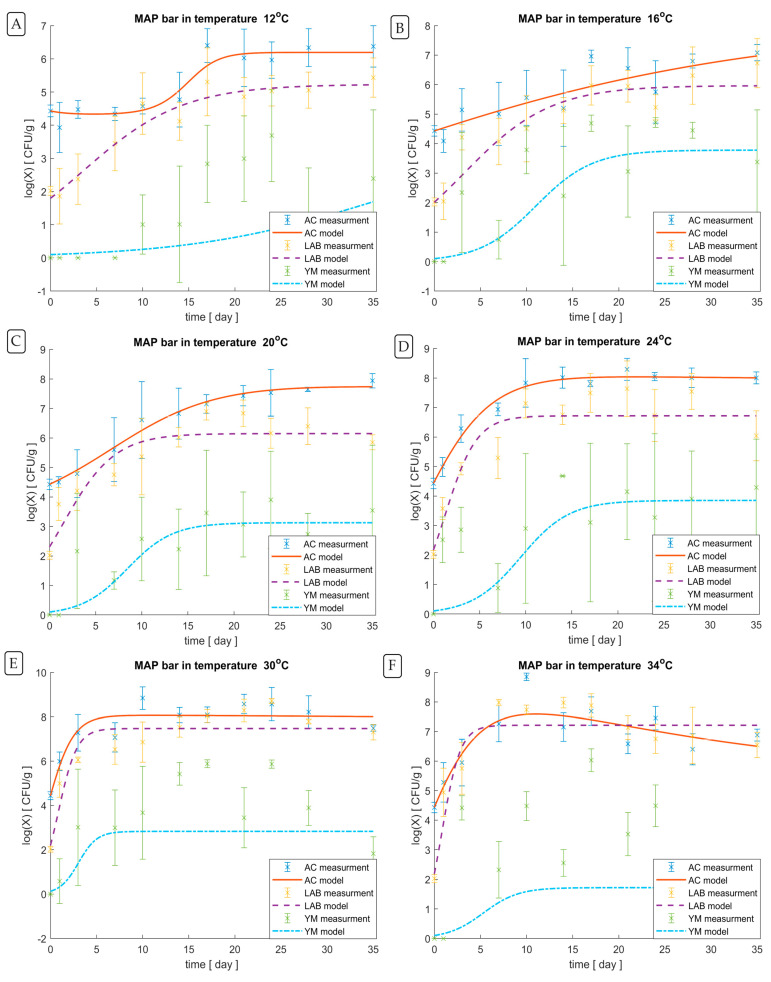
Growth models (the mean count with standard deviation) for APC, LAB, yeast and mould in the MAP_NC bars at 12 (**A**), 16 (**B**), 20 (**C**), 24 (**D**), 30 (**E**), and 34 (**F**) °C during storage up to 35 days (log cfu/g).

**Table 1 foods-12-04372-t001:** Definition of treatments and sampling schemes.

Code	Packing and Storing Conditions	Storage Temperature (°C)	Measurement Day
VP-Con	Vacuum-packaged control (no storing)	-	0
MAP-Con	Modified air-packaged control (no storing)	-	0
VP_C	Vacuum-packaged and refrigeration storage	4, 6, 8	3, 7, 12, 14, 16, 21
MAP_C	Modified air-packaged and refrigeration storage	4, 8	1, 3, 7, 10, 14, 17, 21, 24, 28, 35
VP_NC	Vacuum-packaged and non-refrigeration storage	12, 16, 20, 24	3, 7, 12, 14, 16, 21
MAP_NC	Modified air-packaged and non-refrigeration storage	12, 16, 20, 24, 30, 34	1, 3, 7, 10, 14, 17, 21, 24, 28, 35

**Table 2 foods-12-04372-t002:** Values of the coefficients in the model and the coefficient of determination for the model of APC growth/survival in MAP bars at non-refrigeration temperatures.

Temperature (°C)	Values of Coefficients				R^2^
	r	K	A	B	
12	6.09	12.00	1.82	18.44	0.9188
16	0.08	12.00	6.05	0.31	0.7902
20	0.57	11.86	0.10	1.31	0.9878
24	0.36	9.92	7.58	0.88	0.9828
30	0.81	8.30	7.68	0.30	0.8608
34	0.66	12.00	4.56	1.97	0.8121

r—the growth/death rate; A, B, and K—constant coefficients; R^2^—coefficient of determination.

**Table 3 foods-12-04372-t003:** The values of the coefficients in the model (Equation (11)) and the coefficient of determination for the model of the growth/survival of LAB in the MAP_NC bars.

Temperature (°C)	Values of Coefficients		R^2^
	r	K	
12	0.18	5.23	0.9458
16	0.21	5.97	0.8794
20	0.36	6.15	0.8762
24	0.58	6.72	0.8707
30	1.00	7.47	0.8577
34	1.00	7.22	0.8692

r—the growth/death rate; K—constant coefficient; R^2^—coefficient of determination.

**Table 4 foods-12-04372-t004:** The values of the coefficients in the model (Equation (11)) and the coefficient of determination for the model of the growth/survival of yeast and mould in the MAP_NC bars.

Temperature (°C)	Values of Coefficients		R^2^
	r	K	
12	0.10	3.39	0.3498
16	0.33	3.78	0.6705
20	0.40	3.13	0.7636
24	0.38	3.85	0.5773
30	1.00	2.83	0.5870
34	0.52	1.72	0.2269

r—the growth/death rate; K—constant coefficients; R^2^—coefficient of determination.

## Data Availability

The data presented in this study are available upon request from the corresponding author. The data are not publicly available due to the privacy of sensitive data that may be used by the food processing plant’s competitors, exposing them to economic losses and patenting the product along with the results of its storage tests.

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
