# Peer review of "The Influence of Storage Temperature and Packaging Technology on the Durability of Ready-to-Eat Preservative-Free Meat Bars with Dried Plasma"

_foods, 2023, doi:10.3390/foods12234372_

Round 1
Reviewer 1 Report
Comments and Suggestions for Authors
The scientific research project, in itself, was set up correctly, apart from the use of a VIDAS method for the determination of Salmonella and Listeria which should have been updated to ISO oriami standards applied by almost all laboratories, starting from official laboratories. The topic of the article, however, is not particularly original. Maybe I'm wrong, but it gives the impression that the article was written as a "useful development" of a storage test conducted (like many others) to establish the shelf life of that particular product.
The article is particularly long and within it I have pointed out small oversights that can be improved. In particular, I believe that tables 2 and 3 can be taken outside the text, in the accessory material that can be attached. This would make reading the work itself easier.
Also pay attention to a small typo in the references that I have highlighted like the others in the text I attach.
Aside from a few minor typos, the introduction is well written. I highlighted a paragraph which, in my opinion, is superfluous, makes the introduction too long and adds nothing specific to the topic of the article. See attachment, thanks.
Greetings

Author Response
Dear Reviewer,
Thank you for your very positive response in the first major revision of the article. We hope that we will meet your expectations during the review and revision process. We are submitting a revised version based on all comments from the 3 reviewers in round 1.
We have highlighted in yellow the parts that have been improved from version 1 based on reviews from all three reviewers, including minor and major typos.
We removed the marked 2 paragraphs (lines 76-98 from the version 1) that did not bring anything new to the topic from the Introduction section in accordance with your review.
Indeed moving Tables 2 and 3 to the Supplementary Material will actually improve the readability of the already quite long article manuscript.
Comparison of groups differentiated based on packaging method was carried out only at specific temperatures (e.g. MAP and VP for 4 degrees Celsius) and to determine trends in refrigerated conditions (up to 8 °C) and non-refrigerated conditions (above 8 °C) for each of the packaging methods, to illustrate which method packaging and storage temperature are the most microbiologically optimal for product safety. If you believe that the comparative (temperature) conditions should be unified for both packaging systems in order to clarify the research methodology, we will remove the data for VP bars at 6 °C and for MAP bars at 30 °C and 34 °C.
We have attached two source articles [8, 9] to justify the choice of VIDAS tests to determine the presence of Salmonella and L. monocytogenes. According to the results of the attached articles, the sensitivity and specificity of VIDAS tests in samples with low concentrations of the two pathogens mentioned above are comparable to those of PCR tests and plate cultures.
Yours faithfully,
Authors
Reviewer 2 Report
Comments and Suggestions for Authors
Thank you for having opportunity to review the paper entitled "Influence of storage temperature and packaging technology on the durability of Ready-to-Eat preservative-free meat bar with dried plasma".
Authors did a fair research job, gathered a plethora of data and provided a moderate level of scientific rigor.
After an extensive review, I am of opinion that manuscript could be considered for the publication, however after major revision which would enable novelty in the field.
The main issues with this paper are:
- claims stated in Line 721-725. Namely, in order to claim this as true (that "pork bars do not represent risk") authors should have contaminated respective pork bars with respective food pathogens and measure respective counts' variations. Conducted study is pure form of durability study, not the challenge study required to support claims mentioned above. So, I respectfully suggest authors to reconsider Discussion section.
- the manuscript lacks Conclusions.
- Campylobacter should have been tested according to the ISO and in qualitative mode since the authors referred it as food safety pathogen (Line 74). Moreover, 2073/2005 recognizes Campylobacter in a "process hygiene" criteria, not as "safety".
- in general, majority of references pertaining to the RTE packaging and associated microbial flora are outdated and they should be updated to reflect the most recent state-of-the-art in field of the RTE packaging.
There are also some comments I would like to draw attention to:
Line 79-80. Not necessarily true. RTE meat products are conventionally vacuum packaged, easily re-sealable and do not cause discolorations vs. MAP packaged products.
Line 97-80. I would say this is obsolete nowaday. Majority of RTE FBO's use 7- or 9-layers axial composite polymeric films encompassing EVA. Please, update references.
Line 153-163. This whole section needs to be rephrased and simplified, or moved to discussion section since it introduces some facts rendering introduction readability lower.
Line 189-202. This clearly should be moved to the Materials and Methods section.
Line 215-223. Detailed methodology analytics description is needed here. Please, update.
Line 263. This standard was withdrawn in 2017, and matter was re-assessed and updated by ISO 18787:2017.
Line 266. Second part of the sentence (...and the following units...) is unnecessary, please delete it.
Line 293. Why authors did not quantify L. monocytogenes, since for this type of products it was mandatory (EU Reg. 2073/2005)? Actually, the whole idea is that L. mono count should not cross 100 cfu/g threshold during the shelf life? This was also confirmed by authors in Lines 623-625.
Tables 2 and 3 are hardly readable given the shear size and in my humble opinion should be considered for the graphical representation, including SD, indicators of statistical significance. Same is true for the Table 4.
Line 379-380. Authors wrote P<0.01 and stated that it was not significant. Please, check if this was an error.
Line 630-635. Authors discussed organoleptic changes in pork bars while this was not at all mentioned in the Materials and Methods?
Author Response
Dear Reviewer,
Thank you for your very positive response in the first major revision of the article. We hope that we will meet your expectations during the review and revision process. We are submitting a revised version based on all comments from the 3 reviewers in round 1.
We have highlighted in yellow the parts that have been improved from version 1 based on reviews from all three reviewers, including minor and major typos.
Line 731-735: You are actually right. Our durability test results only inform us that these bacterial species were not present until the 35th day. To confirm the conclusion from version 1 regarding the lack of threat from these specific species of bacteria in this particular product, it is necessary to supplement these studies with challenge tests in which these bars are artificially contaminated (our research team is currently conducting these tests, but their results are planned to be included in the next article). I have revised this paragraph, taking into account your comments - if you think it would be better to remove it from the manuscript, please reply.
The inclusion of Campylobacter spp. in the food safety criteria was indeed an error. Quantification using the ELFA method (e.g. in the TEMPO bioMerieux scheme) has been validated with the standard procedure ISO 10272-1:2006 and can be qualified as an alternative test method. To respectfully support my thesis, I am sending the doi of the article: 10.4081/ijfs.2018.7180. However, if you respectfully think that due to the incorrectly adopted testing methodology for Campylobacter spp., the results and the related discussion should be removed, please reply.
Indeed, most of the references are from over 5 years ago, so based on your comment, we decided to analyze the literature related to alternative methods of food preservation (for paragraph in line 764-768) and added 3 articles describing the most modern methods possible to reflect the most recent state-of-the-art in field of the RTE packaging. Let us know if you wanted to update resources for comparison our results of durability tests with the rest new articles about RTE packaging.
Line 79-80 and 97-100 was deleted due to the comment from other Reviewer. As you can see we are not the food technologies, so that respectfully the material of food packaging is not quite our subject of scientific interest. We assume that specialists in this field have more knowledge on this particular topic.
Line 153-163: This paragraph was moved to the Discussion section and deleted the sentence with parameter A because it did not bring anything new to the topic and only made it more difficult to understand. We hope it looks now more simplified.
Line 189-202: I moved last paragraph from Introduction to Methods and Materials according to your comment.
Line 215-223: Actually it wasn’t analysed from literature nor our research team. The values was prescribed from the product specifications. The trade name and manufacturer of dried plasma used to produce the RTE meat bars was added.
Line 263: Indeed you are right. Mentioned by us standard was outdated. Even though Aqualab 4TE is compatible to ISO 18787:2017, which was mentioned in your comment. I changed the number of standard in manuscript.
Line 293: The intention of the study was to count L. monocytogenes, but due to the extensiveness of the study, qualitative tests were performed using VIDAS. In case of a positive result (L. monocytogenes present in 25 g of sample), the bar sample would be immediately inoculated according to ISO 11290-2:2017 for accurate abundance estimation. Due to the fact that none of the samples were positive (absent at 25 g, which is a more stringent level than below 100 cfu/g), no accurate abundance measurement was performed. We can assume that all samples achieved the result of 0 cfu/g.
Indeed table 2, 3 and 4 are too extensive and disturb the readability of the manuscript. The second Reviewer proposed to move them to Supplementary Material (available for readers), which is in our opinion a good idea. Do you agree, Sir/Madam?
Line 379-380: Indeed there was a typo. Corrected.
Line 630-635: Although the assessment of the organoleptic characteristics of the products was not the main research factor during durability tests, it was impossible not to notice an unpleasant odor and visual changes in the color of the bar's surface and texture when preparing samples for microbiological tests. This paragraph should be treated rather as a digression, hence its inclusion in the Discussion. If you think this is unnecessary, this fragment will of course be removed.
Yours faithfully,
Authors
Reviewer 3 Report
Comments and Suggestions for Authors
Dear authors, although the manuscript is interesting and novel, it is important to address the following points:
Line 7,9,11: remove –
Line 7,9,11: use superscript text format in membership numbering
Line 19: use the non-abbreviated form for both bacteria (E. coli and L. monocytogenes), abbreviations can be used after being mentioned for the first time
Line 20: insert space between the temperature value and the unit °C. It is necessary to modify through the document
Line 21: variance analysis or analysis of variance
Line 25: from which temperature in this study are considered high
Line 31: information is not included in the introductory section
Line 31: Use the correct text format, review the authors' guide (Microsoft Word Template). https://www.mdpi.com/files/word-templates/foods-template.dot
Line 33: 2.1. Product Sample
Line 42: lactate, E327;
Line 42: sodium acetate, E262;
Line 42: dioxide, E551;
Line 47: calcium (108 mg), phosphorus (1530 mg)….. use the same format for the other components
Line 57: What is the meaning of FBO?
Line 57: It is necessary to put the equipment information (model, brand and country) in parentheses. Correct where appropriate in the document
Line 58: Figure 1, it is necessary to change hours to h; insert space between the numerical value and the temperature unit; insert space 5mm; It is necessary to standardize information, in some boxes the text begins with capital letters and in others with lowercase letters. Doubts, is the process mentioned in Figure 1 self-made or did I need to include the reference of the process?
Line 59: 2.2. Storage
Line 65: abbreviations were not previously mentioned in the text
Line 66: use italic text formatting for scientific names
Line 77: use the correct format for tables, vertical lines should not be used
Line 79: use the corresponding indentation format
Line 82,83: S. aureus was omitted?
Line 86: Figure 2, it is necessary to use mL instead of ml throughout the document; min instead of minutes; g instead of grams; µL instead of µl; Regarding Wa, although I understand the meaning in the text, the abbreviation is not previously mentioned
Line 87: 2.3. Physicochemical Examination
Line 88: use the abbreviation of water activity
Line 90: the days do not correspond to those mentioned in the previous table
Line 91: use the abbreviation of water activity
Line 93: include the reference number in square brackets
Line 96: include the reference number in square brackets
Line 97: 2.4. Numerical Analysis of Microbiological Indicators in RTE Meat Bars
Line 99,100: is it necessary to repeat the analysis days if they are previously described in the table?
Line 101: Sometimes TEMPO® appears or as TEMPO, homogenize format throughout the document
Line 107: include vortex equipment information
Line 110: include incubator equipment information
Line 116: Results were
Line 119: It should not be indicated in which tables the results appear in the materials and methods section
Line 97-121: were the references omitted?
Line 122: 2.5. Immunoassay Confirmation of the Presence of L. monocytogenes and Salmonella spp. in Meat Bars
Line 126: Sometimes VIDAS® appears or as VIDAS, homogenize format throughout the document
Line 128: 25 g
Line 230: 225 mL
Line 131: 2 min
Line 135: 25 °C for 26 h
Line 136: 42 °C for 24 h
Line 136: transferred to microtubes to
Line 137: include water bath equipment information
Line 124-145: were the references omitted?
Line 146: the statistical analysis should appear at the end of the section
Line 163: use correct format to cite references in the text
Line 190: 3. Results
Line 191: 3.1. Storage Tests of Refrigerated RTE Meat Bars
Line 195: Regarding the information contained in the table, the information on microorganisms for which there are no results can be eliminated and only mentioned in the text. Also, the results of APC, LAB, S. aureus, Yeast and mold can be changed to graph format, in order to better predict the information
Line 203: CFU/g or cfu/g, homogenize format through the document
Line 206: at 4, 6 and 8 °C, homogenize format through the document
Line 223: (p<0.01) or (P < 0.01), homogenize format through the document
Line 248: 3.2. Storage Tests of RTE Meat Bars Stored under Non-refrigerated Conditions
Line 252-254: standardize text type and size
Line 254: follow the recommendations in the previous table
Line 327: 3.3. Changes of pH and Wa Values in RTE Bars during Storage
Note: Throughout the results section, data that is already in the tables is repeated. This writing style is practical when the information is only presented in figures. It is not recommended to repeat information in the text to reduce its length and improve its understanding.
Note: It is not clear why determinations were made on certain days and not on others.
Note: Although there are many results obtained, it is essential to mention the most important things to reduce the length of the text and improve its understanding.
Line 400: It is recommended to use a layout of figures rather than mentioning them separately
Line 452: use abbreviature for water activity
Line 470: The names of the cited authors should not be mentioned, only the corresponding number should be put in square brackets. correct through document
Reference section: It is necessary to review the correct format to present references (Microsoft Word Template)
Author Response
Dear Reviewer,
Thank you for your very positive response in the first major revision of the article. We hope that we will meet your expectations during the review and revision process. We are submitting a revised version based on all comments from the 3 reviewers in round 1.
We have highlighted in yellow the parts that have been improved from version 1 based on reviews from all three reviewers, including minor and major typos.
Line 25: “Higher temperatures” refer to the previous sentence, which talked about temperatures up to 8 °C. “Higher temperatures” means in this sentence researched temperatures higher than 8 °C, i.e. 12, 16 °C, etc.
Line 31: What do you mean? The whole information in Introduction section was included in manuscript. Based on your line numbering, we assume that you received the editor's version without the Introduction section.
Line 57: FBO means “food business operators”. We used non-abbreviatedly in the Introduction section.
Line 58: We changed “hours” to “h” and gave spaces. We unified every element starting with a capital letter as it is usually done in production schemes based on HACCP. The process in Figure 1 is self-made.
Line 65: The abbreviations were presented earlier in Introduction section – APC means Aerobic Plate Count, LAB means Lactic Acid Bacteria. Escherichia coli, Staphylococcus aureus and Listeria monocytogenes were also presented earlier so that we used abbreviations of their genus name.
Line 82, 83: Indeed S. aureus was omitted accidentally. We moved also Campylobacter spp. to Process Hygiene Criteria.
Line 86: We changed the Figure 2 according to your comment. The abbreviation of Wa (water activity) was used in Introduction section.
Line 90: The measurement of pH and wa was ancillary measurements and wasn’t the main factor examining the influence of temperature and packaging method on the microbiological durability of RTE meat bars so that measurement days of pH and wa don’t fully corespond to measurement days of bacterial counts.
Line 97-121: The protocols for each procedures for each test for the TEMPO system were included in the test kit instructions. If you think they should be included in the References, we will of course do so. However, if you think that the ISO standards under which these tests are equivalent should be cited here, we will do so as well. We just ask that you clarify which references, if they are even needed to be cited here, are more desirable and appropriate here.
Line 126: I unified VIDAS® and TEMPO® through manuscript with the exception in References for not to change the original titles of articles
Line 124-145: Situation similar to the paragraph regarding TEMPO in section 2.5. for line 97-121. Please reply if you would like us to add anything here for References (test kit instructions or rather ISO standards).
We are considering moving Table 2 and Table 3 to Supplementary Material available to readers (as commented by another Reviewer), which would eliminate the issue of repetition in the text through the Results section with detailed means included in the above mentioned tables. What do you think about this idea?
Yours faithfully,
Authors
Round 2
Reviewer 2 Report
Comments and Suggestions for Authors
Thank you for having opportunity to review the revised version of manuscript "Influence of storage temperature and packaging technology on the durability of Ready-to-Eat preservative-free meat bar with dried plasma".
Authors addressed some of the issues raised in the first round of review and thank you for the efforts done in that sense. However, there are still some critical major points unresolved which require authors to revise the manuscript to bring it, at least partially, within the "Foods" Q1 publication standards' framework.
Major issues:
1. Conclusion section is missing. This was clearly indicated in the first round of review and authors should carefully write this section.
2. Aim of the study clearly goes to the end of Introduction section (Line 158-161), so I advise authors to rephrase whole section 2.1 for better readability.
3. Introduction is now well written, but requires condensation of text pertaining to the modelling (from Line 105 to Line 154). It is too long.
4. Sensorial testing issue (Line 604-614). I understood this section is digression. However, if authors have not set organoleptic panel (and they did not, as indicated in the reply to reviewers), this findings bear no scientific rigor, so should be deleted.
Minor issues:
1. Please, move Tables 2 and 3, but also Table 4 into Supplementary Material in order to make manuscript easier to read.
2. Why water activity unit is labeled as Wa, instead of aw (Line 464 onward)?
3. Line 239 (...total number of APC). Terminology is "total count of ...). Please, address this terminology throughout the text.
Author Response
Dear Sir/ Madam,
Thank you for your positive response to the second major revision of the article. We are submitting a revised version based on all comments from the two reviewers (reviewer 1 did not have any comments or did not send them by susy.com) in round 2.
Firstly, we apologize for missing the explanation about the Conclusion section in the previous message. We created Section 5. Conclusions based on the last 4 paragraphs of Section 4. Discussion from the previous version because the Authors' Guide did not require a separate section for conclusions. However, due to the recommendations of both Reviewers, we decided to supplement it.
We have removed section 2.1. General information and aim. We edited some of the sentences from it to the last paragraph of section 1. Introduction as a separation of the main aim of the work. The remaining subsection numbers of section 2. Methods and Materials have been corrected accordingly.
After in-depth analysis, in the opinion of the authors' team, Campylobacter was correctly classified as a food safety criterion (FSC) in the first version of the manuscript. The microbiological criterion constitutes not the pathogen itself but a set of several factors, including, among others, the type of food to which the criterion applies. Initially, Campylobacter was not included in Regulation 2073/2005; this is only being added to the abovementioned regulation in 2017 (Commission Regulation (EU) 2017/1495 of 23 August 2017 amending Regulation EC No 2073/2005 as regards Campylobacter in broiler carcases) and only for Carcases of broilers (fresh meat) as PHC. If ever Campylobacter is introduced into Regulation 2027/2005 for finished processed products, it will serve as an FSC. Similar to what is currently developed for Salomella: Salmonella in fresh carcasses is treated as PHC, and Salmonella in the finished product is treated as FSC, with entirely different limits that are also part of the criterion.
The fragment of the introduction on mathematical modelling has been condensed by approximately 20%.
The digression on the characteristics of the course of organoleptic changes has been removed.
Large tables 2, 3, and 4 from the previous version of the manuscript have been moved to the Supplementary Material as tables S2, S3, and S4. The numbering of tables 5-7 from the previous version has been changed to tables 2-4 in the latest version of the manuscript.
The abbreviation for water activity has been changed throughout the text from Wa to aw.
All expressions regarding the count of microorganisms have been changed from "total number of ..." to "total count of ...".
Following the recommendation of one of the reviewers, the entire manuscript was subjected to an extensive English revision.
Your sincerely,
Authors

Reviewer 3 Report
Comments and Suggestions for Authors
Dear authors, although the manuscript is interesting and novel, it is important that the points described in the Comments and suggestions for authors section be addressed.
Line 31: modify text format, use 1. Introduction instead of 1. INTRODUCTION
Line 42: did you mean 3 L?
Line 45: rewrite….0.2%, and fat 0.2%; while mean mineral…and chloride 2,923 ppm. In addition, a mean of 7.8 for pH values is reported [3].
Line 68: remove space [6,7].
Line 69: The RTE, high-protein
Line 77: rewrite… 4 °C
Line 78: insert the meaning of MAP
Line 80 : remove space [12,13].
Line 80: rewrite… A previous report says that up to…L. monocytogenes [14]. Note: According to the author guide, it is necessary to avoid the use of surnames or names of the authors in the text, the reference should only be included in number format in square brackets.
Line 82: rewrite… L. monocytogenes [14]. Also, a previous research examined 1049..
Line 85: rewrite… Moreover, it was confirmed that…
Line 86: rewrite… Another study showed that using…
Line 90: rewrite… [17]. Other work compared three….
Line 95: rewrite…. Furthermore, it has been described that meat products…
Line 101: insert the meaning of VP-
Line 112: insert abbreviation for water activity (Wa)
Line 118: 1,000 or 1000 like the format described line 46,47 (2,734 or 2734; 2,912 or 2912); it is necessary homogenized the text number format through the manuscript
Line 135: rewrite… It has been used the Gopertz…
Line 137: rewrite… [27]. Also, it has been confirmed that there is…
Line 142: rewrite… consuming RTE egg products… Note: when an abbreviation is used in text it is necessary to use it throughout the document where it is required
Line 150: rewrite… modelling, a previous investigation predicted
Line 151: (cfu) or cfu like in line 152
Line 154: Due to the above, the objective of the work was...'?
Line 155: modify text format, use 2. Materials and Methods instead of 2. METHODS AND MATERIALS
Note: You can consult the text format in the following link https://www.mdpi.com/files/word-templates/foods-template.dot
Line 159: rewrite… packaging technologies (MAP and VP) during….
Line 185: composed of fat 2% (…), carbohydrates 2% (…. Note: homogenize text format like in line 45
Figure 1. Could you include in the figure an image of the final product obtained?
Figure 1. It is necessary to adjust the font size of the figure content to the same as the figure title
Line 209: rewrite… such as Wa and pH…
Line 223: which PHC
Line 225: and FSC
Figure 2. insert space between the number and time., example 72 h, 24 h
Line 227: pH and Wa of RTE…
Lien 230: bars, Wa and pH
Line 233: use Wa instead of wa
Line 245: TEMPO®
Line 251: insert space…. 48 h
Line 251: 24 h
Line 252: 24 h
Line 252: 72 h
Line 253: 24 h
Line 253: 48 h
Line 257: remove (MPN). Note: If the abbreviation will not be used throughout the manuscript, it is not necessary to use it
Line 258: log cfu/g
Line 266: VIDAS®
Line 267: remove space… [8,9].
Line 283: VIDAS®
Note: according to the authors' guide, the numbering of each of the equations must be placed on the right side at the end of the line where it appears.
Line 304: Wa
Line 326: Indicate which factors were considered for the two-way analysis.
Line 326: Although a two-way anova is used, in the results section effects are indicated for either time and temperature, regarding packaging, nothing is described based on the double interaction. Therefore, it is important to include in the tables the P values for each effect and that of the double interaction.
Line 329: modify text format, use 3. Results instead of 1. RESULTS
Note: insert a horizontal line to separate the data that corresponds to each evaluated parameter
Line 341: The APC values in the MAP_C…
Line 345: ANOVA showed
Line 382: day 3
Line 383: day 21
Line 465: Wa (modify through the manuscript)
Table 4: check the font of the information contained in the table, this should be Palatyno Linotype
Line 476: 3.4. Determination of Mathematical Models of Microbial Growth/Survival in RTE Bars
Table 6,7: remove bold text format for 12
Line 538: Figure 3.
Figure 3: Separate the text from the footer of the figure; insert space between the number and temperature symbol, i.e. 12 °C, 16 °C, etc.; on axis use cfu/g instead of CFU/g
Line 544: 14th
Line 584: modify text format, use 3. Discussion instead of 1. DISCUSSION
Note: Do not use the author's name or surname in the text, only the reference number in square brackets should be used.
Line 590: Wa
Line 593: Wa
Line 620: for the VP group
Line 621: for the MAP group
Line 640: VP technology
Line 643: LAB in VP products
Line 645: at 4 °C VP is
Line 646,647: 30th
Line 652: VP RTE
Line 679: insert spaces… 4.3 log.. 3.2 log.. 1.5 log… 1.8 log
Line 698 VP group
Line 699: MAP group
Line 701: MAP group
Line 700: VP group
Line 701: MAP group
Line 728: low-O2
Line 742: use [45–59] instead of [45-59]
Line 772: PHC
Line 774: VP technology
Line 779: VP bars
Line 791: use italic text format for scientific names
Line 794: Conclusion section was omitted?
Reference section
Line 819: Packaging concepts for ready-to-eat food: Recent progress.
Line 819: use the abbreviated form of the journal of the reference used
Line 820: 2017, 1(3), 113–126. https://doi.org/10.1007/s41783-017-0019-9
Line 822: use the abbreviated form of the journal of the reference used
Line 822: 2011, 1, 477–482. https://doi.org/10.1016/j.profoo.2011.09.073
Line 823: Studies on slaughter animal blood plasma: I. Composition of bovine and porcine plasma.
Line 824: use the abbreviated form of the journal of the reference used
Line 824: 1978, 2, 31–38.
Line 828: 2011, 65, 466–475.
Line 833: use the abbreviated form of the journal of the reference used (Note: correct through this section)
Line 833: 2023, 86, 100013. https://doi.org/10.1016/j.jfp.2022.11.005
Line 835: detection
Line 835: variety of foods
Line 836: environmental samples: Collaborative study
Line 836: 2013, 96, 808–821. https://doi.org/10.5740/jaoacint.CS2013_01
Line 839: 2008, 80, 43–65. https://doi.org/10.1016/j.meatsci.2008.05.028
Line 840: complete authors reference, see link https://efsa.onlinelibrary.wiley.com/doi/abs/10.2903/sp.efsa.2023.EN-7827
Line 841: 2023, 20, 7827E. https://doi.org/10.2903/sp.efsa.2023.EN-7827
Line 843: 2015, 55, 103–114. https://doi.org/10.1016/j.foodcont.2015.02.037
Line 844: 2010, 16, 16–23. https://doi.org/10.1111/j.1469-0691.2009.03109.x
Line 847: 2018, 12, 104–112. https://doi.org/10.1007/s11684-017-0593-9
Line 850: 2014, 36, 212–216. https://doi.org/10.1016/j.foodcont.2013.08.035
Line 856: use italic text format for scientific names
Line 852: 2014, 26, 261–267.
Line 855: 2022, 379, 109843. https://doi.org/10.1016/j.ijfoodmicro.2022.109843
Line 856: Little, C.L.; Sagoo, S.K.; Gillespie, I.A.;…
Line 856: level… other
Line 856: use italic text format for scientific names
Line 857: species in selected retail ready-to-eat foods
Line 857: 2009, 72, 1869–1877. https://doi.org/10.4315/0362-028X-72.9.1869
Line 859: J.D.; Hall, P.A.
Line 859: use the correct text format for cited chapter books
Line 862: 2022, 46, 100834. https://doi.org/10.1016/j.cofs.2022.100834
Line 864: 1–10
Line 866: 2013, 29, 451–460. https://doi.org/10.1016/j.foodcont.2012.05.048
Line 867: F.I.; Hellingwerf, K.J.; Teixeira de Mattos, J.M.
Line 868: 2008, 128, 16–21. https://doi.org/10.1016/j.ijfoodmicro.2008.04.029
Line 870: 2018, 57, 229–243. https://doi.org/10.21307/PM-2018.57.3.229
Line 871: predictive model, developed under isothermal conditions
Line 872: broth, to predict growth in ground beef during cooling
Line 872: 2004, 70, 2728–2733. https://doi.org/10.1128/AEM.70.5.2728-2733.2004
Line 875: 2015, 45, 290–299. https://doi.org/10.1016/j.fm.2014.06.026
Line 877: 2021, 53, 657–668. https://doi.org/10.9721/KJFST.2021.53.5.657
Line 878: microbial risk
Line 879: assessment for Campylobacter jejuni in ground meat products
Line 879: 2019, 39, 565–575. https://doi.org/10.5851%2Fkosfa.2019.e39
Line 882: 2020, 118, 107421. https://doi.org/10.1016/j.foodcont.2020.107421
Line 884: aerobic plate count
Line 885: beef surface using fluorescence fingerprint
Line 885: 2013, 7, 1496–1504. https://doi.org/10.1007/s11947-013-1167-8
Line 887: Mason, I.G.
Line 890: 2018, 91, 113–122. https://doi.org/10.1016/j.foodcont.2018.03.027
Line 892: 2010, 101, 8158–8165. https://doi.org/10.1016/j.biortech.2010.06.009
Line 894: 2014, 23, 25–32. https://doi.org/10.1016/j.ifset.2014.03.003
Line 896: 1999, 52, 299–305. https://doi.org/10.1016/S0309-1740(99)00006-6
Line 898: G.D.; de Aragão, G.M.; García-Gimeno, R.M.
Line 900: 2013, 48, 2580–2587. https://doi.org/10.1111/ijfs.12252
Line 901: M.G.
Line 901: The stability and shelf life of meat and poultry
Line 901: use correct text format for chapter books
Line 904: 2005, 100, 253–260. https://doi.org/10.1016/j.ijfoodmicro.2004.10.024
Line 906: 2005, 22, 505–512. https://doi.org/10.1016/j.fm.2005.01.003
Line 909: processing environment of cured meat products
Line 909: 2023, 12, 2161. https://doi.org/10.3390/foods12112161
Line 910: F.T.; El Jabri, M.; Le Page, J.F.;
Line 911: 2015, 148, 43–52. https://doi.org/10.1016/j.jfoodeng.2014.09.040
Line 912: H.M.;
Line 915: Kim, H.W.; Lee, K.; Kim, S.H.; Rhee, M.S.
Line 916: 2018, 70, 129–136. https://doi.org/10.1016/j.fm.2017.09.017
Line 917: M.E.;
Line 918: 2003, 20, 561–566. https://doi.org/10.1016/S0740-0020(02)00154-5
Line 920: 2012, 154, 107–112. https://doi.org/10.1016/j.ijfoodmicro.2011.02.027
Line 922: K.G.;
Line 922: C.W.
Line 924: 2011, 12, 407–415. https://doi.org/10.1016/j.ifset.2011.07.009
Line 925: D.G.; Farkas, D.F.
Line 926: 1989, 54, 1547–1549. https://doi.org/10.1111/j.1365-2621.1989.tb05156.x
Line 927: I.S.;
Line 928: 2008, 49, 68–112. https://doi.org/10.1080/10408390701764278
Line 930: Microbial control by high pressure processing for shelf-life extension of packed meat products
Line 931: the cold chain: Modeling and case studies
Line 931: 2021, 11, 1317. https://doi.org/10.3390/app11031317
Line 934: 2021, 12, 275–281. https://doi.org/10.1016/j.ifset.2011.04.006
Line 935: J.H.
Line 936: , 3, 113–126. https://doi.org/10.1016/S1466-8564(02)00012-7
Line 937: B.M.; Carballido, J.R.;
Line 937: P.R.
Line 939: 2019, 120, 38–51. https://doi.org/10.1016/j.foodres.2019.02.025
Line 940: D.S.
Line 940: Antimicrobial packaging for meat products
Line 940: use correct text format for chapter books
Line 942: N.E.; Roura, S.I.
Line 942: effectiveness of bioactive packaging materials
Line 943: edible chitosan and casein polymers
Line 943: carrot, cheese, and salami
Line 943: 2010, 76, M54–M63. https://doi.org/10.1111/j.1750-3841.2010.01910.x
Line 946: 2014, 178, 7–12. https://doi.org/10.1016/j.ijfoodmicro.2014.02.013
Line 948: 2019, 5, 324–346. https://doi.org/10.3934%2Fmicrobiol.2019.4.324
Line 951: 2023, 35, 100997. https://doi.org/10.1016/j.fpsl.2022.100997
Line 952: Innovative food packaging, food quality and safety, and consumer perspectives
Line 953: 2022, 10, 747. https://doi.org/10.3390/pr10040747
Line 955: Current trends of food analysis, safety, and packaging
Line 955: 2021, Article ID 9924667. https://doi.org/10.1155/2021/9924667
Line 928: 2016, 121, 253–260. https://doi.org/10.1016/j.meatsci.2016.06.021
Line 960: D.L.;
Line 961: 2020, 137, 109645. https://doi.org/10.1016/j.foodres.2020.109645
Line 964: 2018, 38, e12493. https://doi.org/10.1111/jfs.12493
Line 965: Holman, B.W.B.; Kerry, J.P.; Hopkins, D.L.
Line 966: 2018, 144, 159–168. https://doi.org/10.1016/j.meatsci.2018.04.026
Author Response
Dear Sir/ Madam,
Thank you for your positive response to the second major revision of the article. We are submitting a revised version based on all comments from the two reviewers (reviewer 1 did not have any comments or did not send them by susy.com) in round 2.
After in-depth analysis, in the opinion of the authors' team, Campylobacter was correctly classified as a food safety criterion (FSC) in the first version of the manuscript. The microbiological criterion constitutes not the pathogen itself but a set of several factors, including, among others, the type of food to which the criterion applies. Initially, Campylobacter was not included in Regulation 2073/2005; this is only being added to the abovementioned regulation in 2017 (Commission Regulation (EU) 2017/1495 of 23 August 2017 amending Regulation EC No 2073/2005 as regards Campylobacter in broiler carcases) and only for Carcases of broilers (fresh meat) as PHC. If ever Campylobacter is introduced into Regulation 2027/2005 for finished processed products, it will serve as an FSC. Similar to what is currently developed for Salomella: Salmonella in fresh carcasses is treated as PHC, and Salmonella in the finished product is treated as FSC, with entirely different limits that are also part of the criterion.
Large tables 2, 3, and 4 from the previous version of the manuscript have been moved to the Supplementary Material as tables S2, S3, and S4. The numbering of tables 5-7 from the previous version has been changed to tables 2-4 in the latest version of the manuscript.
We created Section 5. Conclusions based on the last 4 paragraphs of Section 4. Discussion from the previous version because the Authors' Guide did not require a separate section for conclusions. However, due to the recommendations of both Reviewers, we decided to supplement it.
Line 31: corrected to 1. Introduction instead of 1. INTRODUCTION
Line 42: corrected from 3 l to 3 L
Line 45: rewritten to "(..) and fat 0.2%; while mean mineral (…) and chloride 2,923 ppm. In addition, a mean of 7.8 for pH values is reported [3]."
Line 68, 80, 267: removed spaces
Line 69: used the abbreviation of RTE
Line 77: rewritten to 4 °C
Line 78: inserted "modified atmosphere packaging"
Line 80: rewritten to "A previous report says that up to…L. monocytogenes [14]."
Line 82: rewritten to "L. monocytogenes [14]. Also, previous research examined 1049"
Line 85: rewritten to "Moreover, it was confirmed that"
Line 86: rewritten to "Another study showed that using"
Line 90: rewritten to "[17]. Other work compared three"
Line 95: rewritten to "Furthermore, it has been described that meat products"
Line 101: inserted "(vacuum packaging)"
Line 112: inserted abbreviation for water activity (aw)
Line 118: homogenized the text number format through the manuscript to 1,000 from 1000
Line 135: rewritten to "It has been used the Gompertz"
Line 137: rewritten to "[27]. Also, it has been confirmed that there is"
Line 142: rewritten to "consuming RTE egg products"
Line 150: rewritten "modelling, a previous investigation predicted"
Line 151, 258: homogenized the format of "cfu/g" throughout the whole text
Line 154: added the aim of the work "Due to the above, the objective of the work was...", paraphrasing the 2.1 General information and aim from the previous version of the manuscript, which was deleted and the next subsection numbers were sequentially moved from 2.2. on 2.1. etc.
Line 155: modified to 2. Materials and Methods
Line 159: rewritten to "packaging technologies (MAP and VP) during"
Line 185: homogenized to "composed of fat 2% (including saturated fatty acids 0.8%), carbohydrates 2% (including sugars 1%), protein 74.9% and salt 11.9%."
Figure 1. homogenized the font size in the figure content. To include the image of the final product obtained, we have to contact the manufacturer due to the lack of any pictures of the above.
Line 209, 227, 230, 233, 304, 465, 590, 593: The abbreviation of water activity was homogenized throughout the whole text to "aw"
Line 223, 225, 772: used abbreviation of PHC and FSC
Figure 2., Line 251-253 inserted spaces before "h"
Line 245: changed to "TEMPO®"
Line 257: removed (MPN).
Line 266, 283: changed to "VIDAS®"
Line 326: An error was made in specifying the methodology used - we used a single, not a double, ANOVA, hence the presentation of the analysis data in the form of effects separately for time and temperature in Table S2 and S3. Changed two-way ANOVA to one-way ANOVA
Line 329: changed to "3. Results"
Line 341: changed to "The APC values"
Line 345: changed analysis of variance to ANOVA
Line 382: changed 3rd day to day 3; unified the form of notation throughout the manuscript
Line 383: changed 21st day to day 21
Table 6,7: removed bold text format for 12
Line 544, 646,647: used superscript
Line 584: modified text format to "4. Discussion"
Line 620, 640, 643, 645, 652, 698, 700, 774, 779: used the abbreviation of "VP"
Line 621, 699, 701: used the abbreviation of MAP
Line 679: inserted spaces before log
Line 728: used O2 abbreviation
Line 742: used [45–59] instead of [45-59]
Line 791: used italic text format for L. monocytogenes
Reference section: abbreviated journal names used; re-edited book chapters with Author's Guide guidelines; the form of doi administration was unified; spaces between the initials of the authors of the publication have been removed.
Following your recommendations, we have edited the fragments in which the names of the authors of the cited publications appeared throughout the text.
Following the recommendation of one of the reviewers, the entire manuscript was subjected to an extensive English revision.
Yours sincerely,
Authors
